# Study protocol for a randomized controlled trial of a group-adapted Somatic Experiencing® intervention for Indonesian women survivors of sexual assault with PTSD symptoms

Ligina Ayudia[1]*, Fredrick Dermawan Purba[2,4], Annemarie Samuels[3], Aulia Iskandarsyah[2,4]

1 Postgraduate Program in Psychology, Faculty of Psychology, Universitas Padjadjaran, Bandung, Indonesia, 2 Department of Psychology, Faculty of Psychology, Universitas Padjadjaran, Bandung, Indonesia, 3 Institute of Cultural Anthropology and Development Sociology, Leiden University, Leiden, The Netherlands, 4 Center for Psychological Innovation and Research, Faculty of Psychology, Universitas Padjadjaran, Bandung, Indonesia

* ligina18001@mail.unpad.ac.id

## Abstract

### Background

Sexual assault affects 35.6% of women globally. In Indonesia, 1 in 3 women aged 15–64 have experienced physical and/or sexual assault. This often leads to post-traumatic stress disorder (PTSD) and related symptoms. The standard PTSD treatments have cultural and resource limitations, which emphasize the need for a culturally adapted intervention. Somatic Experiencing® (SE)®, a body-oriented trauma therapy, has shown promise in reducing PTSD symptoms across diverse populations, but its cultural applicability and effectiveness in Indonesia remain under-investigated. This study addresses that gap by adapting SE®, for Indonesian women survivors of sexual assault.

### Methods

This study is a parallel-group, randomized controlled trial (RCT) designed to assess the effectiveness of a 10-session group-based SE® intervention, compared withcontrol group. A total of 207 participants will be recruited and randomly assigned using a 2:1 allocation ratio (n = 138 intervention, n = 69 control). The intervention consists of structured group activities based on SE® principles, targeting improvements in PTSD symptoms, resilience, and quality of life. Furthermore, participants will be assessed at multiple points using standardized measures (PCL-5, CD-RISC-25, WHOQOL-BREF). Analysis will be conducted using SPSS version 22.0 for quantitative data and thematic coding for qualitative insights.

**Data availability statement:** All relevant data from this study will be made available upon study completion. Deidentified research data will be made publicly available through OSF when the study is completed and published. OSF registration link: https://osf.io/a8x7d Project DOI: https://doi.org/10.17605/OSF.IO/A8X7D.

**Funding:** This study is supported by Padjadjaran University Research Grant (No. 1549/UN6.3.1/PT.00/2023). The funding body has no role in study design, data collection, or reporting. There was no additional external funding received for this study.

**Competing interests:** The authors have declared that no competing interests exist.

**List of Abbreviations:** ANS: Autonomic Nervous System; CBT: Cognitive Behavioural Therapy; CD-RISC-25: Connor-Davidson Resilience Scale (25-item version); CONSORT: Consolidated Standards of Reporting Trials;DSM-5: Diagnostic and Statistical Manual of Mental Disorders, 5th Edition; HAT: Helpful Aspects of Therapy; ITT: Intention-to-Treat; NGO: Non-Governmental Organization; PCL-5: PTSD Checklist for DSM-5; PPA: Pusat Pelayanan Terpadu (Integrated Service Center); PTSD: Post-Traumatic Stress Disorder; RCT: Randomized Controlled Trial;SCID: Structured Clinical Interview for DSM Disorders; SE: Somatic Experiencing®; SIBAM: Sensation, Image, Behaviour, Affect, Meaning; SRS: Session Rating Scale; TAU: Treatment as Usual; WHOQOL-BREF: WHO Quality of Life–Brief.

## Discussion

This study represents the first RCT of culturally adapted SE® intervention in Indonesia. The findings are expected to inform trauma-focused clinical practice in low-resource settings and contribute to the global understanding of body-based therapy effectiveness in diverse cultural contexts. Results may also provide evidence for scalable group interventions targeting PTSD among women survivors of gender-based violence.

## Trial registration number

ISRCTN58257113 (Registration date: 29 October 2024)

## Trial sponsor

Universitas Padjadjaran, Faculty of Psychology, Jl. Raya Bandung-Sumedang K21, Jatinangor 45363, Indonesia.

## Introduction

Sexual assault affects 35.6% of women globally. In Indonesia, 1 in 3 women aged 15-64 have reported having experienced physical and/or sexual assault during their lifetime, either by a partner or a non partner [1]. A 2016 online survey by Lentera Sintas Indonesia showed that 46.7% of 25,213 participants had experienced some form of sexual assault [2]. Additionally, Komnas Perempuan (2021) recorded 955 cases of sexual assault in public and neighborhood spaces during 2020 [3].

Sexual assault refers to any attempt to obtain sexual acts through coercion or force, such as rape by an intimate partner or strangers. Epidemiological studies have shown that sexual assault is the trauma with the highest risk of resulting in post-traumatic stress disorder (PTSD) [4]. Neurobiologically, PTSD often emerges from dysregulation of autonomic responses from incomplete or overwhelmed defensive reactions to trauma. These can manifest as hyperarousal (e.g., panic, hypervigilance), hypoarousal (numbness, fatigue), or dissociative states (e.g., detachment, disconnection). During traumatic events, the autonomic nervous system (ANS) may enter an emergency alarm state, characterized by activation of sympathetic fight-or-flight responses or parasympathetic shutdown (freeze or collapse), depending on the individual's neurobiological and contextual factors. When these defensive responses remain incomplete such as an urge to run or resist that is never carried out, they may become ''*stuck*'' in the body, contributing to chronic stress and PTSD symptoms [5,6]. It is important to note that not all survivors respond in the same way; dissociation, emotional numbing, and immobilization may occur with or without classical ''*freeze*'' responses. Although this response may provide temporary protection, it causes unexpressed energy to accumulate in the body. Over time, this unresolved activation disrupts the hypothalamic-pituitary-adrenal (HPA) axis, leading to demoralization, interpersonal dysfunction, stigma, and diminished self-regulation capacity [7,8].

Despite growing access to legal and psychological support, many women with trauma histories face barriers to care, including cost, stigma, lack of trauma-informed providers, and low treatment retention [9]. Although evidence-based treatments such as eye movement desensitization and reprocessing (EMDR), Cognitive Behavioral Therapy (CBT), and Prolonged Exposure (PE) have shown efficacy in reducing PTSD symptoms [10], dropout rates can reach 18%, largely due to difficulties tolerating trauma exposure and autonomic dysregulation [11–13].

Top-down approaches often depend heavily on verbal expression, memory, and cognitive processing. These demands can exceed the capacities of individuals facing trauma-related impairment in language, attention, and executive function, particularly in low-resource contexts like Indonesia, where access to mental health care is scarce, limited cognitive capacity, and executive functioning limitations may impede engagement in cognitively demanding treatment [3,14]. These limitations highlight the need for bottom-up approaches. Recent neurobiological models, such as the Polyvagal Theory emphasizes the importance of addressing a wide spectrum of autonomic responses in trauma, supporting methods that directly target interoceptive and proprioceptive processes to regulate the nervous system and restore safety [15].

Somatic Experiencing® is a biopsychological model developed by Levine (1998). The term ''*soma*'' refers to the living, sensing body, while ''*experiencing*'' emphasized being present in the here and now. SE® facilitates completion of defensive body reaction (biological survival responses), stabilizes dysregulated nervous system activity, and enhances resilience through titrated exposure to body-based cues of trauma.

SE® targets core physiological regulation by engaging the autonomic nervous system-both the sympathetic and para-sympathetic branches through interoception (visceral sensations), proprioception (body awareness), and kinesthetic feedback (movement), helping survivors of sexual assault learn to tolerate, integrate, and transform stress responses through (i) orienting, which helps survivors re-establish safety by gently directing attention to the present environment, counteracting hypervigilance and dissociation; (ii) felt sense, cultivates awareness of subtle internal sensations, enabling survivors of sexual assault who often experience emotional numbing or disconnection from the body to gradually reconnect with embodied experience; (iii) resourcing provides access to internal or external anchors of safety, strengthening the nervous system's capacity to regulate distress; (iv) titration involves approaching traumatic material in small, manageable increments, which prevents retraumatization in survivors who may be easily overwhelmed by intrusive memories; (v) pendulation guides the oscillation between states of activation and calm, supporting autonomic flexibility and reducing chronic immobility; (vi) finally, discharging allows incomplete defensive responses to be released through small motor movements or physiological shifts, helping survivors resolve the ''trapped'' survival energy characteristic of sexual trauma. Together, these mechanisms enable sexual assault survivors to restore autonomic balance, reclaim bodily agency, and rebuild resilience, offering structured ways to reconnect with bodily signals, thereby improving both mental and physical health [16–18].

Group-based SE® provides additional co-regulatory benefits that can be especially valuable in collectivist cultures like Indonesia, where healing through shared experience and low-verbal processing aligns with social norms and resource limitations. It also offers potential advantages, including cost-effectiveness, normalization of trauma symptoms, and peer support, especially in collectivist cultures like Indonesia. Social support has also been shown to buffer PTSD symptoms and improve emotional regulation.

The primary aim of this trial is to evaluate the clinical effectiveness of a culturally adapted group-based SE®intervention. Feasibility and cultural adaptation were conducted in prior stages and are not within the scope of the current trial [19,20]. The hypothesis states that intervention can reduce PTSD symptoms, improve resilience, and foster quality of life for participants, providing an accessible, evidence-based method that respects Indonesia's cultural context.

## Materials and methods

### Study design

This study is a parallel-group, randomized controlled trial (RCT) designed to evaluate the effectiveness of a culturally adapted, group-based SE® intervention for Indonesian women survivors of sexual assault with PTSD symptoms. The trial

adheres to the SPIRIT 2013 guidelines and is registered at ISRCTN (ISRCTN58257113). The SPIRIT schedule is presented in Fig 1.

Ethical approval was obtained from the Health Research Ethics Committee of Padjadjaran University (No. 1357/UN6.KEP/EC/2023) on 16 November 2023. Written informed consent will be obtained from all participants prior to data collection. Recruitment is scheduled to start in January 2026 and end by June 2026. Written informed consent will be obtained from all participants prior to data collection. Consent will be in written form, signed by participants, and securely stored.

| | STUDY PERIOD | | | | | | |
|---|---|---|---|---|---|---|---|
| | Enrollment | Allocation | Post-randomization | | Close-out | | |
| TIMEPOINT | 6 months | 7 days | Week-1 – week-5 | Week-6 – Week-10 | T1 | T2 | T3 |
| **ENROLMENT** | | | | | | | |
| **Eligibility Screen** | X | | | | | | |
| **Informed consent** | X | | | | | | |
| **Baseline Assessment** | X | | | | | | |
| **Allocation** | | X | | | | | |
| **Interventions** | | | | | | | |
| **Active treatment (3 hours)** | | | ———————————————————————————— | | | | |
| **Treatment as usual (90 minutes)** | | | ————————— | | ————— | | |
| **Assessments:** | | | | | | | |
| Age, marital status, education level, occupation | X | | | | | | |
| SCID-5 | X | | | | | | |
| HAT | | | ———————————————— | | | | |
| SRS | | | ———————————————— | | | | |
| PCL-5 | | X | | | ————— | | |
| CD-RIS-25 | | | | | | | |
| WHOQoL-BREF | | X | | | ————————— | | |
| | | X | | | ————————— | | |

SCID (Structured Clinical Interview for the DSM-5); TAU (Treatment As Usual (Counselling with Cognitive approach)); PCL-5 (Posttraumatic Stress Disorder Checklist for DSM-5Checklist); CD-RIS-25 (The Connor-Davidson Resilience Scale); WHOQoL-BREF (The brief version of World Health Organization Quality of Life); HAT (Helpful Aspects of therapy form); SAT (Satisfaction Aspects of therapy); SRS (Session Rating Scale)

**Fig 1. Participant timeline schedule of enrollment in accordance to SPIRIT 2013 guidelines.**

## Timeline

Recruitment is planned to begin in January 2026 and end by June 2026 in Bandung, West Java. The intervention is expected to be delivered over a 3-month period, with data analysis projected for completion in 2028. All study procedures will be documented according to CONSORT and SPIRIT reporting standards.

## Participants

### Eligibility criteria.
Inclusion criteria:

- Female, aged 18 -45 years

- A self-reported history of sexual assault

- PTSD symptoms in the mild to moderate range (PCL-5 score ≥ 39)

- Trauma occurred ≥ 6 months prior to enrollment (to ensure symptom stability and chronicity)

- Not currently receiving trauma-focused psychological treatment or medication within the last 30 days

  Exclusion criteria:

- Substance or alcohol misuse

- Ongoing severe mental illness (e.g., psychosis, bipolar disorder, organic brain disorders)

- PTSD symptoms arising in conjunction with a diagnosed medical condition

- Current suicidal ideation requiring emergency care

## Recruitment and setting

Participants will be recruited from multiple settings, including psychology bureaus, women's support organizations, hospitals, and foundations. Recruitment will also take place through online platforms (Instagram, Twitter/X, WhatsApp, LINE). Eligibility will be confirmed through initial screening via phone or in-person, followed by clinical assessment using SCID-5 modules adapted for the Indonesian population. Recruitment is expected to take place from January to June 2026. Written informed consent will be collected before any data collection or participation in sessions. Consent will be documented using a standardized consent form approved by the ethics committee.

To minimize participant attrition, several retention strategies will be implemented: (i) flexible scheduling options for group sessions; (ii) transportation subsidies for eligible participants; (iii) weekly reminders sent via the participant's preferred contact method (e.g.,WhatsApp or SMS); (iv) emotional check-ins and pre-session grounding will be conducted in person at the beginning of each session, guided by SE® practitioners to support autonomic regulation and participant safety. These practices help orient participants to the present moment and reduce pre-session anxiety. For participants who miss a session, a follow-up emotional check-in will be offered via WhatsApp or SMS to maintain continuity of care and reduce dropout; (v) proactive communication one day prior to each session, including an outline of the next session's agenda and a short message asking if any support is needed. This message will be sent via WhatsApp or SMS to foster engagement and address potential barriers early.

## Randomization and Blinding

Randomization will be conducted by an independent data coordinator using Castor EDC. Group allocation will be concealed from clinical assessors but not from the study coordinator or group facilitators. Blinding of participants is not feasible due to the nature of the intervention. A recruitment and retention flowchart is based on CONSORT guidelines as presented in Fig 2.

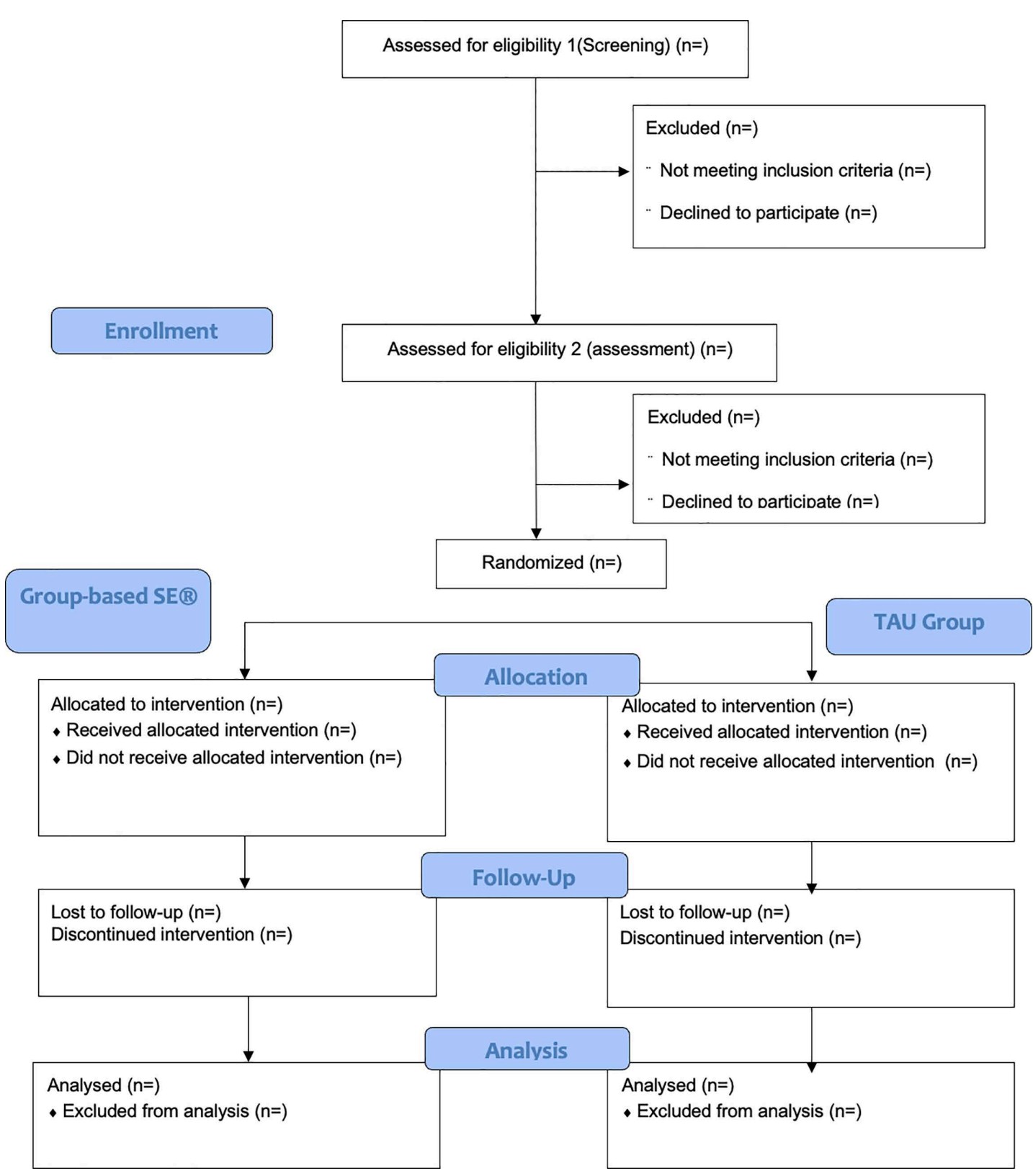

**Fig 2. Recruitment and retention flowchart based on CONSORT guidelines.**

## Intervention

**Group-based SE® Intervention.** Participants in the intervention arm will attend 10 weekly group sessions (3 hours/session) of manualized, group-based SE® intervention based on SE® principles. The intervention emphasizes bottom-up processing of trauma, stabilization of the autonomic nervous system, and completion of incomplete defensive responses. A Module Structure of Group-based SE® sessions is presented in Table 1.

**Discontinuation criteria.** Participants may discontinue participation at any time upon their own request, or if the facilitator identifies significant emotional distress requiring individual therapy. In such cases, participants will be referred for appropriate psychological support. As the intervention is non-pharmacological, no modification of dosage or session content will be applied.

## Concomitant care

Participants may continue receiving any ongoing counselling or medical care throughout the trial. However, initiation of new trauma-focused therapy during trial participation is discouraged to avoid potential confounding effects on the study outcomes.

## Ancillary and post-trial care

Participants who experience psychological distress related to the intervention or trial participation will be offered post-trial psychological support and referral to university counselling services. No financial compensation is provided, but free psychological follow-up will be available on request.

## Intervention fidelity and monitoring

Each group-based SE® session will be delivered by trained SE® practitioners and supervised by a senior clinical psychologist. At the end of each session, participants will complete:

- **Session Rating Scale (SRS)** for therapeutic alliance
- **Helpful Aspects of Therapy (HAT)** form for qualitative feedback

Grounding exercises will be conducted before and after each session to promote emotional regulation.

## Treatment As Usual (TAU)

Participants in the control arm will receive five group counselling sessions (90minutes per session) over five weeks. These sessions focus on psychoeducational counseling session, covering basic mental health support strategies and coping techniques, and are facilitated by licensed psychologist. The decision to provide five 90 minute per sessions is based on evidence from Indonesian research and practice, which showing that survivors of sexual assault typically receive between 4–7 session of psychological support(Ayudia et al. 2025c; Sumekar and Sunawan 2019).

**Outcome Measures and Timing.** Outcomes will be measured at four time points:

- **T0**: Baseline (Week 0)
- **T1**: Post-treatment (Week 10)
- **T2**: 1-month follow-up (Week 14)
- **T3**: 3-month follow-up (Week 22)

Table 1. Module Structure of Group-based SE® sessions.

| No | Module Title | Content/Activities | Intention | Target Issues | Skill Taught | Homework | Supplies |
|---|---|---|---|---|---|---|---|
| 1 | Preparatory 1 | • Welcome and Introduction<br>• Facilitator greeting, acknowledging courage to attend<br>• Icebreaker: each participant states their name and one personal resource<br>• Building cohesion and safety<br>• Orientation exercise<br>• Group agreement:<br>• Psycho-education: Introduce SE® and PTSD<br>• Brief explanation of trauma as incomplete defensive responses<br>• Importance of body-based approaches and group support.<br>• How SE® group can help<br>• Experiential Exercise:<br>• Guided practice<br>• Invite sharing<br>• Normalize varied responses<br>• Integration and Group Agenda<br>• Reflection<br>• Collect and record group goals<br>• Emphasize collaborative ownership process<br>• Closing<br>• Summarize today's agenda<br>• Introduce homework<br>• Closing ritual: each participant shares one word/gesture for how they leave the session | • Build group cohesion and establish a sense of safety.<br>• Orient participants to group purpose, structure and rules.<br>• Introduce SE® and its relevance to PTSD recovery<br>• Facilitate gentle re-connection to bodily sensation and resources | • Relational safety and trust.<br>• Disconnection from the body and avoidance of interoception<br>• Uncertainty about group role, expectations, and confidentiality<br>• Normalization and universality (reducing isolation). | • Grounding and orienting in the present moment.<br>• Identifying and using internal and external resources.<br>• Sensation tracking (pleasant, neutral, unpleasant).<br>• Respectful communication and group process skills. | | • Name cards<br>• Sensation name cards<br>• SE® PowerPoint |
| 2 | Preparatory 2: Cultivating embodied connection and safety | • Orienting<br>• Psycho-education (overview trauma)<br>• Introduce SE® skills<br>• Resourcing practice<br>• Integration and reflection | • Deepen orienting practices and reinforce safety in the here-and-now.<br>• Provide an overview of trauma and PTSD (psycho-education).<br>• Introduce SE® skills in more detail (e.g., pendulation, titration).<br>• Develop personal and relational resources to strengthen inner stability.<br>• Encourage group reflection to consolidate learning. | • Limited sense of control: Trauma often brings helplessness; resourcing fosters empowerment.<br>• Body avoidance: Survivors may resist noticing bodily sensations; gradual titration is essential.<br>• Connection and universality: Seeing others share similar challenges supports hope and belonging | • Orienting: Noticing external environment and sensory details to create safety.<br>• Resource building: Identifying supportive people, places, memories, or strengths that bring comfort.<br>• Basic SE skill introduction: Pendulation (shifting between comfort/discomfort) and titration (breaking down overwhelming experience into small pieces).<br>• Reflective sharing: Articulating sensations and insights in simple, safe language. | • Resource practice: Each day, recall or visualize one supportive person, place, or memory. Notice body sensations connected to it.<br>• Orienting practice: Spend 1–2 minutes daily naming 3 external details in the environment when feeling tense. | • SE® PowerPoint<br>• Resourcing worksheet |

*(Continued)*

**Table 1.** (Continued)

| No | Module Title | Content/Activities | Intention | Target Issues | Skill Taught | Homework | Supplies |
|---|---|---|---|---|---|---|---|
| 3 | Tracking 1 | •Orienting<br>•Felt sense<br>•Psycho- education window of tolerance<br>•Sympathetic & parasympathetic arousal cycle exercise<br>•Reflection | •Strengthen group safety through orienting and embodied awareness.<br>•Introduce and practice felt sense as a key SE skill.<br>•Provide psycho education on the Window of Tolerance.<br>•Explore sympathetic vs. parasympathetic arousal cycles through gentle exercises.<br>•Foster reflection and integration of learning. | •Survivors may experience dysregulated arousal without understanding why → psycho education builds clarity and reduces fear.<br>•Many have difficulty accessing body sensations safely → felt sense expands capacity for awareness.<br>•Misinterpretation of physiological arousal (e.g., anxiety=danger) → normalizing sympathetic/parasympathetic shifts reduces shame.<br>•Universality and instillation of hope: realizing that others experience similar regulation challenges. | •Orienting to present environment as a stabilizing anchor.<br>•Felt sense: cultivating subtle awareness of inner bodily experience.<br>•Window of Tolerance awareness: recognizing when inside vs. outside the optimal arousal zone.<br>•Tracking arousal cycles: noticing activation (sympathetic) and settling (parasympathetic).<br>•Reflective group sharing to strengthen integration. | •Felt sense practice: take 2–3 minutes daily to notice inner sensations without judgment.<br>•Window tracking: journal one moment/day noticing whether you were inside, above, or below the window of tolerance. | • Sensation and emotion name cards |
| 4 | Tracking 2 | •Orienting<br>•Felt sense<br>•Screening type of boundaries<br>•Setting boundaries exercise<br>•Projecting voices<br>•Reflection | •Reinforce safety through orienting and felt sense practice.<br>•Introduce the concept of boundaries and how trauma disrupts them.<br>•Support participants in exploring their personal boundary styles (rigid, porous, flexible).<br>•Practice boundary-setting exercises in a safe, contained group format.<br>•Use voice projection as an embodied way of reclaiming power and self-expression.<br>•Foster reflection to consolidate learning and promote universality. | •Many survivors experience blurred or violated boundaries → psycho education and exercises rebuild awareness and choice.<br>•Difficulty saying "no" or expressing needs → voice projection strengthens agency.<br>•Shame or fear about self-assertion → normalization through group sharing reduces isolation (Yalom: universality, instillation of hope).<br>•Disconnection from bodily cues of safety/danger → felt sense deepens awareness of limits. | •Orienting & felt sense as preparation for boundary work.<br>•Identifying boundary styles: rigid, porous, flexible.<br>•Practical boundary-setting: saying no, using hand gestures, maintaining physical space.<br>•Voice projection: practicing tone and volume to embody agency.<br>•Reflective group sharing for integration and mutual support. | •Boundary journal: notice one situation daily where you felt your boundaries respected or challenged.<br>•Voice practice: at home, practice saying "no" or "stop" aloud in a grounded stance.<br>•Felt sense reflection: track sensations when setting or thinking about boundaries. | •SE®PowerPoint<br>•Tail<br>•Tissue |

*(Continued)*

**Table 1.** (Continued)

| No | Module Title | Content/Activities | Intention | Target Issues | Skill Taught | Homework | Supplies |
|---|---|---|---|---|---|---|---|
| 5 | Tracking 3 | •Orienting<br>•Felt sense<br>•Resourcing<br>•Pendulation<br>•Titration<br>•Reflection | •Reinforce safety and presence through orienting and felt sense.<br>•Strengthen personal stability by deepening resourcing.<br>•Introduce pendulation (moving between discomfort and comfort) to expand capacity for regulation.<br>•Introduce titration (working with small amounts of activation) to prevent overwhelm.<br>•Support reflection and normalization through group sharing. | •Survivors often swing between hyperarousal and hypoarousal→pendulation builds tolerance and flexibility.<br>•Fear of overwhelm when sensing trauma-related activation→titration introduces safe pacing.<br>•Difficulty staying embodied→resourcing anchors awareness.<br>•Shame about reactions→universality and group reflection reduce isolation | •Orienting & felt sense as preparatory anchors.<br>•Resourcing: strengthening inner/outer supports.<br>•Pendulation: shifting gently between activation and settling.<br>•Titration: working with small, tolerable bits of activation.<br>•Reflective practice: verbalizing sensations, noticing change. | •Pendulation practice: notice one mild discomfort in the body, then shift focus to a pleasant/neutral sensation.<br>•Titration journaling: practice recalling a very small piece of a stressful cue, then return to resource — note sensations and responses<br>•Daily orienting: 1–2 minutes naming safe details in the environment. | •SE® PowerPoint<br>•Ball |
| 6 | Discharging 1 | •Orienting<br>•Felt sense<br>•Psycho- education: Emotion as Messengers<br>•Resourcing<br>•Titration<br>•Pendulation<br>•Discharging | •Reinforce safety and grounding through orienting and felt sense<br>•Learn emotions as adaptive signals guiding action and meaning<br>•Support participants in learning to listen to emotions through the body, not only cognitively.<br>•Strengthen resourcing before moving into more challenging material.<br>•Deepen practice of titration and pendulation.<br>•Introduce the concept of discharge as the body's way of releasing trapped survival energy.<br>•Support normalization of discharge experiences in the group. | •Many survivors suppress or fear emotions due to past trauma -> psycho- education reframe emotions as intelligent signals, not threats.<br>•Many survivors often fear body reactions (trembling, shaking, heat) → psycho education reframes these as signs of recovery<br>•Risk of overwhelm when arousal builds→titration and pendulation prevent retraumatization.<br>•Shame or confusion about bodily release→group context fosters universality and acceptance (Yalom: universality, catharsis).<br>•Difficulty noticing small shifts→practice enhances interoceptive awareness. | •Orienting & felt sense as stabilizers.<br>•Resourcing to anchor safety.<br>•Listening to emotions: identifying emotion, noticing where it lives in the body, and sensing its message.<br>•Titration & Pendulation: working with small bits of activation and moving between states.<br>•Recognizing and allowing discharge: trembling, yawning, sighing, warmth, tears, or subtle release.<br>•Group reflection to validate and normalize the process. | •Listening to emotions: identifying emotion, noticing where it lives in the body, and sensing its message.<br>•Discharge awareness: during daily life, notice any small releases (sigh, yawns, warmth, trembling) and journal them.<br>•Pendulation practice: choose one mild discomfort, then return to resource, noting any shifts.<br>•Daily orienting: spend 1–2 minutes naming safe details in the environment. | •SE®PowerPoint |

*(Continued)*

| No | Module Title | Content/Activities | Intention | Target Issues | Skill Taught | Homework | Supplies |
|----|----|----|----|----|----|----|----|
| 7 | Discharging 2 | •Orienting<br>•Felt sense<br>•Psycho- education: Befriending Trauma sensations & Understanding Aggression.<br>•Resourcing<br>•Titration<br>•Pendulation<br>•Discharging | •Strengthen grounding and presence through orienting and felt sense.<br>•Offer psycho education on befriending trauma sensations—developing curiosity and compassion toward body experiences rather than avoidance.<br>•Support participants in understanding aggression as protective life energy, not destructive force.<br>•Deepen capacity for resourcing, titration, pendulation, and discharge.<br>•Normalize bodily responses and reinforce agency through safe somatic exploration. | •Survivors often fear body sensations related to trauma (e.g., tightness, trembling, numbness).<br>•Befriending sensations reduces fear and avoidance.<br>•Misunderstanding aggression as dangerous → re-framing it as protective restores empowerment.<br>•Chronic suppression of body cues → psycho education and group normalization reestablish trust in somatic experience.<br>•Yalom's therapeutic factors (universality, interpersonal learning, catharsis) enhance connection and safety during this deeper work. | •Orienting and felt sense to anchor in safety.<br>•Befriending sensations: observing body experience with curiosity and without judgment.<br>•Recognizing aggression as mobilizing, life-affirming energy.<br>•Resourcing, titration, pendulation to regulate activation.<br>•Allowing discharge as the body's natural completion of activation. | •Befriending practice: once a day, pause and notice one body sensation. Name it, breathe with it, and observe if it shifts.<br>•Aggression awareness: notice any moments of natural strength or "no" energy; track sensations safely.<br>•Pendulation journal: note experiences of moving between activation and calm, and any discharge noticed. | •SE® PowerPoint |
| 8 | Discharging 3 | •Orienting<br>•Felt sense<br>•Resourcing<br>•Titration<br>•Pendulation<br>•Guided reflection<br>•Discharging | •Strengthen grounding and body awareness through orienting, felt sense, and resourcing.<br>•Deepen capacity for titration, pendulation, and discharge.<br>•Support reflection on *how safety and reward have been wired through avoidance versus authentic presence.*<br>•Facilitate insight into how the body experiences relief — whether from escape or connection.<br>•Encourage self-compassion and curiosity toward new patterns of embodied safety and reward. | •Many survivors develop *avoidance-based safety:* feeling secure only when unseen, silent, or conflict-free.<br>•These patterns may suppress self-expression and reinforce chronic freeze/dissociation.<br>•The goal is to bring awareness — not judgment — to these strategies and gently explore alternatives based on presence and connection.<br>•Yalom's factors (self-understanding, universality, interpersonal learning) support self reflection and relational healing. | •Orienting & felt sense to establish present-moment safety.<br>•Resourcing to anchor before emotional reflection.<br>•Titration & pendulation to manage arousal when exploring avoidance patterns.<br>•Tracking discharge as signs of release and new learning.<br>•Somatic self-inquiry: noticing body sensations linked to avoidance vs. connection. | •Presence vs. Avoidance Journal: Each day, note one situation where you felt safe. Reflect—was it from avoiding something, or from connecting/present-moment engagement?<br>•Body awareness check-in: 1–2 minutes daily to sense what safety feels like in your body.<br>•Pendulation practice: move gently between moments of mild discomfort and sensations of ease, noting what supports presence. | •SE®PowerPoint |

*(Continued)*

**Table 1.** (Continued)

| No | Module Title | Content/Activities | Intention | Target Issues | Skill Taught | Homework | Supplies |
|---|---|---|---|---|---|---|---|
| 9 | Returning to equilibrium | • Orienting<br>• Felt sense<br>• Resourcing<br>• Titration<br>• Pendulation<br>• Discharging | • Consolidate and integrate all previously learned SE skills (orienting, felt sense, resourcing, titration, and discharge). • Support participants in meeting new or unfamiliar feelings with presence rather than avoidance. • Normalize that new sensations or emotions may arise as healing deepens. • Encourage curiosity, flexibility, and compassion in facing change. • Celebrate progress and prepare participants for continued self-regulation beyond the group. | • Survivors may fear new sensations, emotions, or desires that emerge as the body "wakes up." • The nervous system may equate novelty with threat; this session re-frames new experiences as opportunities for integration and vitality. • Closure can evoke mixed feelings (sadness, pride, loss); group reflection supports regulation and meaning-making. • Yalom's therapeutic factors (cohesion, interpersonal learning, existential meaning) are central in this final phase. | • Orienting: anchoring in the here-and-now as new experiences arise. • Felt sense: noticing subtle sensations connected to new feelings. • Resourcing: grounding in supportive internal/external anchors. • Titration & pendulation: moving safely between curiosity and comfort. • Discharging: allowing the body to release activation naturally. • Self-reflection: recognizing growth and new relational capacities. | • Daily body check-in: 2–3 minutes noticing new or familiar sensations, meeting them with curiosity. • Self-reflection journal: "What new emotions or sensations did I notice this week? How did I respond?" • Resource reminder: Create a list of personal grounding practices for post-group self-support. • Optional: write a short letter to yourself expressing appreciation for your body's resilience and progress. | • SE® PowerPoint |
| 10 | Returning to equilibrium & Appreciation Post Test | • Opening & Orientation<br>• Settling into Body: Guide participant to sense their breath, contact with support surfaces and invite noticing any signs of equilibrium (ease, calm, balance). • Reflection (Identifying learning): invite participants to reflect on what they have learned over the past weeks. • Orientation: Moving from internal to External: Reinforce appreciation for self and others. • Conduct post-group evaluation (post-test) to measure perceived changes and outcomes. • Closing ritual: group grounding (orient to space), closing circle (each participant shares one word or gesture representing gratitude or equilibrium), facilitator expresses appreciation, normalizes post group emotions, and offers resources for continued support. | • Facilitate transition from internal, trauma-focused work to external, relational engagement. • Encourage reflection on personal growth and learning from the group process. • Reinforce appreciation for self and others. • Conduct post-group evaluation (post-test) to measure perceived changes and outcomes. • Provide structured and emotionally safe closure. | • Survivors may experience anxiety about ending the group or re-entering daily social contexts. • Integration is essential: connecting inner regulation skills to real-world interactions. • Mixed feelings about closure (loss, gratitude, fear of ending) can be acknowledged and regulated. Yalom's therapeutic factors—cohesion, universality, and existential meaning—guide the group through healthy closure. | • Orienting: shifting from internal bodily focus to awareness of others and environment. • Settling: returning to baseline and recognizing signs of equilibrium. • Social engagement: applying regulation skills while connecting externally. • Appreciation and reflection: integrating insights and achievements. • Self-regulation for transition beyond group setting. | • Continued self-care: practice daily orienting and resourcing in daily environments. • Reflection journal: write about ongoing sensations of safety, vitality, or connection. • Connection exercise: identify one supportive person or activity to maintain engagement post-group. • Optional: create a *personal grounding plan* for times of stress or overwhelm. | SE® PowerPoint |

**Primary outcomes include:**

- PTSD Checklist for DSM-5 (PCL-5): Primary outcome measure assessing PTSD symptoms. Indonesian version Cronbach's α = 0.93

**Secondary outcomes include:**

- Connor-Davidson Resilience Scale (CD-RISC-25)

- WHO Quality of Life-BREF (WHOQOL-BREF): Assesses physical, psychological, social, and environmental quality of life. Cronbach's α range: 0.41–0.77

**Other measures include:**

- Sociodemographic questionnaire: Collects age, marital status, occupation, education, and medical history.

- Structured Clinical Interview for DSM-5 (SCID-5): Used for diagnostic confirmation of PTSD and exclusion criteria

## Sample size

A priori power analysis using G*Power 3.1 estimated a required total sample size of 207 participants (138 intervention, 69 control) to detect a medium effect size (Cohen's d = 0.5), with α = 0.05, power = 0.80, and accounting for 30% attrition.

The choice of medium effect size (d = 0.5) is based on prior meta-analytical findings from studies evaluating SE® and other body-oriented trauma interventions among individual PTSD or complex trauma symptoms. Specifically, Payne et al [7] and Andersen et al [16–18] report medium to large effect sizes in PTSD symptom reduction following SE® informed interventions. Given the cultural adaptation and group-based format in this study, a conservative medium effect size was deemed appropriate.

Participants will be randomized in a 2:1 allocation ratio using block randomization generated through Castor EDC software.

## Data collection and management

**Data collection personnel.** Data collection will be carried out by trained research assistants who are postgraduate psychology students with prior experience in research. These assistants will receive additional training on ethical handling sensitive data, trauma-informed interviewing, and scoring procedures for standardized instruments. Data collectors will be blinded to group allocation to minimize bias.

**Participant retention and follow-up.** To promote participant retention and ensure complete follow-up, weekly contact and reminder messages will be provided through phone and WhatsApp by research assistants. For participants who discontinue or deviate from intervention protocols, outcome data from all completed assessments will be retained for intention-to-treat (ITT) analysis.

**Platform and tools used.** All outcome measures, including PCL-5, CD-RISC-25, WHOQOL-BREF, SRS, HAT will be administered using secure digital forms hosted on REDCap (Research Electronic Data Capture), a HIPAA-compliant, encrypted online platform. In cases where participants have limited internet access or prefer paper-based administrations, hard-copy versions will be made available and later digitized into the REDCap system by data entry personnel.

**Confidentiality.** All participant data will be de-identified prior to analysis. Electronic data will be stored on a secure, encrypted university server accessible only to authorized research personnel. Hard-copy documents will be stored in locked cabinets within the Department of Clinical Psychology. Data will be retained for five years after publication, after which it will be permanently deleted. Only the principal investigator and data analyst will have full access to the de-identified dataset

## Statistical analysis plan

All statistical analyses will be performed using SPSS version 22.0.

- Descriptive statistics will be used to summarize demographic and clinical characteristics of the participants.

- Baseline differences between groups will be examined using chi-square test for categorical variables and t-test for continuous variables.

- To evaluate treatment effects, linear mixed-effect models will be used to analyse the interaction of time (T0, T1, T2, T3) and group condition (intervention vs control) on primary and secondary outcomes.

- The models will include fixed effects for time, group, and time x group interaction, and random intercepts for participants to account for within subject variability across time points.

- To minimize confounding related to unequal treatment exposure, total professional contact time (minutes) within and outside the study will be logged for each participant. In prespecified secondary analyses, this variable will be included as a covariate in the mixed-effects models. Additional sensitivity analyses will be conducted, including (i) exclusion of participants who receive substantial trauma-focused services outside the protocol, (ii) per-protocol analyses restricted to participants with adequate adherence (≥ 80% of assigned sessions), and (iii) exploratory dose-response models testing the interactions between group assignment and contact time. These steps will allow us to assess whether intervention effects remain robust after adjusting for variation in professional contact time.

- Effect sizes will be reported using Cohen's d.

- All analyses will follow the intention-to-treat (ITT) principle, including all randomized participants. Missing data will be handled using regression imputation under the assumption that data are missing at random (MAR). Specifically, predictive mean matching will be used to impute missing continuous outcome variables, which is considered appropriate for longitudinal clinical trial data.

   To control for the increased risk of Type I error due to multiple comparisons across outcomes, A Bonferroni correction will be applied.

## Monitoring

As this is a minimal risk behavioral trial, no formal data monitoring committee (DMC) will be established. Oversight will be provided by a senior clinical psychologist independent from the intervention facilitators. No interim analyses are planned. The trial may be stopped early by the principal investigator or the ethics committee in the event of safety or ethical concerns.

   SE® practitioners will monitor participant safety through pre- and post-session grounding, and report any adverse events to the university ethics board. Internal auditing will be conducted every three months by the Department of Clinical Psychology's research ethics monitoring team, which operates independently from the investigators.

## Auditing

Internal auditing will be performed every three months by the Department of Clinical Psychology's research ethics monitoring team. The process is independent from the investigators and sponsor to ensure compliance with ethical and methodological standards.

## Discussion

This trial represents the first RCT of culturally adapted, group-based SE® intervention for Indonesian women survivors of sexual assault. It addresses a critical treatment gap by targeting bottom-up regulation of trauma responses in a low-resource and collectivist context. The group format offers additional benefit of cost-effectiveness, normalization of trauma

symptoms, and peer-based co regulation, which are congruent with Indonesian cultural values. Findings will contribute to the broader literature on culturally adapted trauma interventions, which remains limited in Southeast Asia.

If effective, this body-based group intervention could serve as a scalable trauma recovery model in low-resource and collectivist cultural contexts. The study also seeks to contribute to the broader literature on cultural adaptation of trauma therapies, which has been underexplored in Southeast Asia. Findings may inform mental health policy and service delivery, especially for gender-based violence survivors lacking access to individualized therapy.

### Limitations

This study has several limitations. First, despite efforts to balance treatment exposure, a discrepancy in contact time remains between the SE® intervention (10 sessions x 180 minutes) and the TAU arm (5 sessions x 90 minutes). This discrepancy introducing bias. However, the design of TAU reflects the real-world availability of services in Indonesia, where survivors of sexual assault typically have limited and sporadic access to structured trauma-focused therapy. To address this issue, will be committed to controlling for total minutes of professional contact as a covariate in secondary analyses. Sensitivity analyses will also be conducted to test whether outcomes remain robust after adjusting for contact time.

Second, while the number of session differs between arms, it is important to emphasize that the aim of this trial is not to determine which treatment is superior, but rather to evaluates the effectiveness of culturally adapted SE® intervention relative to the benchmark of usual care in Indonesia. In this sense, TAU serves as a pragmatic comparator that mirrors local service provision, rather than a dose-matched control condition. This design enhances ecological validity and translational relevance, though it limits strict comparability of treatment "dose."

Third, TAU is based on psychoeducational counselling rather than trauma-specific intervention. While this may reduce comparability in therapeutic content, it pragmatically reflects current service delivery in Indonesia, thus increasing ecological validity and translational relevance.

Fourth, blinding of participants is not feasible due to the nature of the intervention, which may increase the risk of expectancy effects. Finally, the trial is conducted within one cultural and geographical setting, which may limit generalizability to other populations. Nonetheless, focus on cultural adaptation offers insight that could guide implementation in similar low-resource and collectivist context.

## Conclusions

This trial aims to generate evidence on the clinical effectiveness of a group-based SE® intervention that has been culturally adapted for Indonesian survivors of sexual assault with PTSD symptoms. If successful, the study could inform service delivery and policy in low-resource settings and strengthen global knowledge on the role of body-oriented trauma therapies.

## Dissemination plans

The findings of this study will be disseminated through multiple channels. First, the primary results will be submitted for publication in peer-reviewed international journal focusing on trauma, global mental health, and psychotherapy. Findings will also be presented at national and international conferences related to psychology, psychiatry, and public health.

To ensure practical impact, dissemination will extend beyond academic forums. Result will be shared with Indonesian stakeholders, including mental health professionals, non-governmental organization (NGOs) supporting women survivors of gender-based violence, and community organizations engaged in trauma recovery and women's health.

Moreover, accessible formats such as policy briefs, podcasts, and community discussions will be used to reach a wider audience including survivors, caregivers, and advocacy groups. This strategy is intended to maximize the potential for clinical translation, inform culturally appropriate trauma care practices in low-resource settings, and support the scalability of group-based SE® intervention in Indonesia.

The full protocol, anonymized dataset, and statistical code will be shared publicly via the Open Science Framework (OSF) repository.

## Acknowledgments

We thank all potential collaborators and trauma support centers for future participation.

## Author contributions

**Conceptualization:** Ligina Ayudia, Aulia Iskandarsyah.

**Data curation:** Ligina Ayudia.

**Formal analysis:** Ligina Ayudia.

**Funding acquisition:** Ligina Ayudia, Aulia Iskandarsyah.

**Investigation:** Ligina Ayudia.

**Methodology:** Ligina Ayudia.

**Project administration:** Ligina Ayudia.

**Resources:** Ligina Ayudia.

**Software:** Ligina Ayudia.

**Supervision:** Ligina Ayudia, Fredrick Dermawan Purba, Annemarie Samuels, Aulia Iskandarsyah.

**Validation:** Ligina Ayudia.

**Visualization:** Ligina Ayudia.

**Writing – original draft:** Ligina Ayudia.

**Writing – review & editing:** Ligina Ayudia, Fredrick Dermawan Purba, Annemarie Samuels, Aulia Iskandarsyah.

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
