## [Decision Letter · Decision Letter 0]

11 Sep 2025

Dear Dr. Ayudia,

Thank you for submitting your manuscript to PLOS ONE. After careful consideration, we feel that it has merit but does not fully meet PLOS ONE’s publication criteria as it currently stands. Therefore, we invite you to submit a revised version of the manuscript that addresses the points raised during the review process.

reply to the reviewers' comments

We look forward to receiving your revised manuscript.

Kind regards,

Fadwa Alhalaiqa

Academic Editor

PLOS ONE

**Journal Requirements:**

1. When submitting your revision, we need you to address these additional requirements. Please ensure that your manuscript meets PLOS ONE's style requirements, including those for file naming. The PLOS ONE style templates can be found at https://journals.plos.org/plosone/s/file?id=wjVg/PLOSOne_formatting_sample_main_body.pdf and https://journals.plos.org/plosone/s/file?id=ba62/PLOSOne_formatting_sample_title_authors_affiliations.pdf 2. Thank you for stating in your Funding Statement: This study is supported by Padjadjaran University Research Grant (No. 1549/UN6.3.1/PT.00/2023). The funding body has no role in study design, data collection, or reporting.  Please provide an amended statement that declares *all* the funding or sources of support (whether external or internal to your organization) received during this study, as detailed online in our guide for authors at http://journals.plos.org/plosone/s/submit-now.  Please also include the statement “There was no additional external funding received for this study.” in your updated Funding Statement. Please include your amended Funding Statement within your cover letter. We will change the online submission form on your behalf. 3. In the online submission form, you indicated that your data will be submitted to a repository upon acceptance.  We strongly recommend all authors deposit their data before acceptance, as the process can be lengthy and hold up publication timelines. Please note that, though access restrictions are acceptable now, your entire minimal  dataset will need to be made freely accessible if your manuscript is accepted for publication. This policy applies to all data except where public deposition would breach compliance with the protocol approved by your research ethics board. If you are unable to adhere to our open data policy, please kindly revise your statement to explain your reasoning and we will seek the editor's input on an exemption. 4. Your ethics statement should only appear in the Methods section of your manuscript. If your ethics statement is written in any section besides the Methods, please delete it from any other section. 5. We note that you have referenced (Almasyhur AF NMW) which has currently not yet been accepted for publication. Please remove this from your References and amend this to state in the body of your manuscript: (Almasyhur AF NMW [Unpublished]) as detailed online in our guide for authorshttp://journals.plos.org/plosone/s/submission-guidelines#loc-reference-style 6. If the reviewer comments include a recommendation to cite specific previously published works, please review and evaluate these publications to determine whether they are relevant and should be cited. There is no requirement to cite these works unless the editor has indicated otherwise. 

Reviewers' comments:

**Comments to the Author**

1. Does the manuscript provide a valid rationale for the proposed study, with clearly identified and justified research questions?

Reviewer #1: Yes

Reviewer #2: Yes

2. Is the protocol technically sound and planned in a manner that will lead to a meaningful outcome and allow testing the stated hypotheses?

Reviewer #1: Partly

Reviewer #2: Partly

3. Is the methodology feasible and described in sufficient detail to allow the work to be replicable?

Reviewer #1: No

Reviewer #2: No

4. Have the authors described where all data underlying the findings will be made available when the study is complete?

Reviewer #1: Yes

Reviewer #2: Yes

5. Is the manuscript presented in an intelligible fashion and written in standard English?

Reviewer #1: Yes

Reviewer #2: Yes

You may also provide optional suggestions and comments to authors that they might find helpful in planning their study.

**Reviewer #1: ** This is an important simple parallel design in which participants will be randomized in a controlled setting (RCT) designed to assess the effectiveness of a 10-session group based Somatic Experiencing intervention, compared to a control group. The design is well planned and the strategy for treatment delivery appears to be reasonable.

This report focuses on the statistical approach. There are several gaps which require clarity.

1. The investigators have to reconcile the ‘Measures and Assessment Points’ in the protocol with Figure 1. For example, in the text the outcomes are:

T0: Baseline

T1: Post-treatment

T2: 1-week follow-up

T3: 4-week follow-up

On Figure 1. There is no 4 week follow-up. Is it week 5? What is week 10 closeout? What is end of Session ( is that post 10 session intervention?). This has to be very clear as to what assessments are given where for the statistical modeling. Please match the T's in the text to the columns on Figure 1 and state clearly the modeling (noted below) times input.

2. The analysis section needs a rewrite and to be made clearer. Is the regression imputation assumed missing at random? What imputation method is reasonable for this type of data?

3. One needs a better rationale for medium effect size (Cohen’s d = 0.5). How was this derived or where was it derived from?

4. Please expand on: ‘Linear mixed models which will analyze the interaction of time and group condition on primary and secondary outcomes’. This makes sense but what are the random and fixed effects? Will there be random intercept models for the three primary outcomes?

5. The ‘Discussion’ section needs a ‘strengths and limitations’ paragraph.

6. Even though there are accommodations made in the sample size calculation for attrition, one needs a recruitment retention strategy to minimize the expected loss.

**Reviewer #2: ** Dear authors congratulations on your protocol, it will be important to test a group approach with SE for women that suffered from sexual assault.

I have some concerns and hope you can accept some suggestions and elucidate some doubts on your protocol before acceptance. I also think they can help improving the study before starting it.

Background : 5th paragraph

I don’t think it is clear when you provide the hypothesis of incomplete stress response to justify using SE for the treatment, I think readers with less experience in PTSD cannot understand the explanation you gave. I think this whole paragraph should be better written and more theory grounded.

You generalized in your affirmation of the freezing response in victims of sexual assault, although it is very frequently to observe dissociative symptoms, a better foundation of your treatment proposal is necessary.

Methods:

Why include adolescents? I think it is better to include >18yo , because you will have more generalizable data with other researchers and more chance to publish, also your age range is extense, menopausal women also tend to have different stress responses. Think if it feasible to have a narrow age range.

Your inclusion period seems to short for the sample size you need, because many women that suffer sexual assault avoid to accept treatment, because of shame, guilty, and avoidance of the traumatic memories, I would recommend to increase that time.

How long since the sexual assault happened? Because chronic symptoms are more difficult to respond. Have you thought about that?

For the sample size many studies show attrition rates for different treatments of almost 30%, but you decided to calculate with 20%, maybe the sample size will lead to no significant result.

The TAU that you mention as a comparison for the 10 SE sessions does not seem adequate, because is it an only 90 minutes session? and that is all the treatment they will receive for PTSD? The amount of time spend with the patients can be a serious confounding factor.

Also I would suggest that you include some instrument to evaluate dissociation that I think is a very frequent symptom for sexual assault victims.

**Do you want your identity to be public for this peer review?** For information about this choice, including consent withdrawal, please see our Privacy Policy

Reviewer #1: No

Reviewer #2: **Yes: ** Andrea Feijo Mello

---

## [Author Response · Author response to Decision Letter 1]

18 Oct 2025

Ms. Fadwa Alhalaiqa

Academic Editor

PLOS ONE

Dear Ms. Alhalaiqa,

Thank you for your valuable feedback and the opportunity to revise and resubmit our manuscript titled “Study Protocol for a Randomized Controlled Trial of Group-Adapted Somatic Experiencing Intervention for Indonesian Women Survivors of Sexual Assault with PTSD Symptoms (PONE-D-25-39124).” We sincerely appreciate the constructive comments from you and the reviewers, which have greatly contributed to enhancing the quality of our work. In response, we have carefully revised the manuscript and prepared a detailed point-by-point response to all reviewer comments.

Editorial Correction

We have ensured that the manuscript met PLOS ONE's style requirements

This study is supported by Padjadjaran University Research Grant (No. 1549/UN6.3.1/PT.00/2023). The funding body has no role in study design, data collection, or reporting.

We have included the statement “There was no additional external funding received for this study.” in our updated Funding Statement. (see page 18)

3. In the online submission form, you indicated that your data will be submitted to a repository upon acceptance. We strongly recommend all authors deposit their data before acceptance, as the process can be lengthy and hold up publication timelines.

We have changed the statement that data will be submitted to a repository before acceptance, which was submitted into OSF, which currently is awaiting registration procedures after being pre-registered with the URL of osf.io/7zeqd.

We have edited the ethics statement in the Methods section (Page ?)

5. We note that you have referenced (Almasyhur AF NMW) which has currently not yet been accepted for publication. Please remove this from your References and amend this to state in the body of your manuscript: (Almasyhur AF NMW [Unpublished]) as detailed online in our guide for authors http://journals.plos.org/plosone/s/submission-guidelines#loc-reference-style

We have removed the reference and amended this in the body of our manuscript as suggested by the editor (Page ?)

We have reviewed all previous published work as requested.

Response to Reviewer

Response to Reviewer #1

Reviewer #1 Comment 1:

The investigators have to reconcile the ‘Measures and Assessment Points’ in the protocol with Figure 1. For example, in the text the outcomes are:

T0: Baseline

T1: Post-treatment

T2: 1-week follow-up

T3: 4-week follow-up

On Figure 1. There is no 4 week follow-up. Is it week 5? What is week 10 closeout? What is end of Session (is that post 10 session intervention?). This has to be very clear as to what assessments are given where for the statistical modeling. Please match the T's in the text to the columns on Figure 1 and state clearly the modeling (noted below) times input.

Response:

Thank you for your observation. We have revised Figure 1 to ensure consistency with the text. The timeline now explicitly labels:

T0: Baseline (Week 0),

T1: Post-treatment (Week 10),

T2: 1-week follow-up (Week 11),

T3: 4-week follow-up (Week 14).

We also clarified the meaning of "End of Session" as referring to the completion of the 10th SE group session (week 10), which coincides with T1. The new figure and corresponding explanation in the text now match the modeling inputs and assessment timepoints.

Action Taken:

Figure 1 updated; Text in "Study Timeline" and "Measures and Assessment" sections revised accordingly.

Reviewer #1 Comment 2:

The analysis section needs a rewrite and to be made clearer. Is the regression imputation assumed missing at random? What imputation method is reasonable for this type of data?

Response:

Thank you for this important feedback. We have revised the Analysis section to clarify that missing data will be handled under the assumption of missing at random (MAR). Regression imputation using predictive mean matching will be applied using SPSS v22.0, which is appropriate for longitudinal clinical data with continuous outcomes.

Action Taken:

“Data Analysis” section updated with imputation assumptions and method.

Reviewer #1 Comment 3:

One needs a better rationale for medium effect size (Cohen’s d = 0.5). How was this derived or where was it derived from?

Response:

We appreciate this point. The medium effect size (Cohen’s d = 0.5) was selected based on meta-analytical findings of SE and other body-based trauma interventions in similar populations (e.g., Payne et al., 2015; Andersen et al., 2023). These studies report medium to large effects in PTSD symptom reduction. This rationale is now included in the sample size section.

Action Taken:

Sample Size subsection updated with citations and justification.

Reviewer #1 Comment 4:

Please expand on: ‘Linear mixed models which will analyze the interaction of time and group condition on primary and secondary outcomes’. This makes sense but what are the random and fixed effects? Will there be random intercept models for the three primary outcomes?

Response:

Thank you. We have now specified that linear mixed-effects models will include fixed effects for time, group, and time × group interaction, and random intercepts for each participant to account for within-subject correlation over time. This model will be applied to all three primary outcomes.

Action Taken:

“Data Analysis” section expanded with random/fixed effects model structure.

Reviewer #1 Comment 5:

The ‘Discussion’ section needs a ‘strengths and limitations’ paragraph.

Response:

Thank you for this valuable suggestion. We have now added a dedicated “Strengths and Limitations” paragraph at the end of the Discussion section. This paragraph highlights the main strengths of the trial (e.g., being the first RCT of culturally adapted SE in Indonesia, the use of a group-based format to enhance scalability, and adherence to SPIRIT and CONSORT guidelines), while also acknowledging key limitations (e.g., unequal session dosage between SE and TAU, reliance on psychoeducational counseling rather than trauma-specific TAU, lack of participant blinding, and limited generalizability to female survivors in one cultural context).

Action Taken:

Final paragraph of Discussion revised to include strengths and limitations.

Reviewer #1 Comment 6:

Even though there are accommodations made in the sample size calculation for attrition, one needs a recruitment retention strategy to minimize the expected loss.

Response:

Acknowledged. We have added a new subsection under “Recruitment and Setting” to explain retention strategies, including flexible scheduling, transportation subsidies, emotional check-ins, and reminders via preferred contact methods.

Action Taken:

Retention strategy paragraph added under Methods → Recruitment and Setting.

Response to Reviewer #2

Reviewer #2 Comment 1:

Background: 5th paragraph

I don’t think it is clear when you provide the hypothesis of incomplete stress response to justify using SE for the treatment, I think readers with less experience in PTSD cannot understand the explanation you gave. I think this whole paragraph should be better written and more theory grounded.

Response:

Thank you for this valuable suggestion. We have rewritten the paragraph to include a clearer, theory-based explanation of the “incomplete defensive response” concept. We now reference Levine’s polyvagal-informed model and the autonomic pathways associated with freeze/dissociation, emphasizing the neurobiological rationale behind Somatic Experiencing (SE) and grounding it within trauma theory. We also included citations from van der Kolk (2014) and Dana (2018) for accessibility.

Action Taken:

Background paragraph 5 revised for clarity and theoretical grounding.

Reviewer #2 Comment 2:

You generalized in your affirmation of the freezing response in victims of sexual assault, although it is very frequently to observe dissociative symptoms, a better foundation of your treatment proposal is necessary.

Response:

We agree with this observation and have removed the generalization. Instead, we now state that while many survivors experience dissociation and hypoarousal, there is individual variability in stress responses. We expanded the discussion to include polyvagal theory perspectives and how SE addresses a range of autonomic responses, not just the freeze state.

Action Taken:

Background revised to reflect nuanced view of stress responses and dissociation.

Reviewer #2 Comment 3:

Why include adolescents? I think it is better to include >18yo, because you will have more generalizable data with other researchers and more chance to publish, also your age range is extensive, menopausal women also tend to have different stress responses. Think if it feasible to have a narrow age range.

Response:

Thank you for this valuable suggestion. We agree that a narrower age range will improve the homogeneity of participants’ physiological and psychological stress responses and enhance the generalizability of the findings. Accordingly, we have revised the inclusion criteria to include only adult women aged 18 to 45 years. This range is consistent with previous trauma-related intervention studies (e.g., Michopoulos et al., 2015; Thoma et al., 2018), which indicate that hormonal and neurobiological stress regulation differs significantly in adolescents and peri-/postmenopausal women. The study protocol, ethical approval, and recruitment materials will be updated to reflect this modification.

Action Taken:

The inclusion criteria in the Methods – Participants section have been revised to specify that only women aged 18–45 years will be included.; Methods section updated.

Reviewer #2 Comment 4:

Your inclusion period seems too short for the sample size you need. I would recommend to increase that time.

Response:

Acknowledged. The recruitment timeline has been extended from 3 months to 6 months (January–June 2026) to improve participant reach and reduce pressure on enrollment. This change is reflected in the “Recruitment and Setting” and “Study Timeline” sections.

Action Taken:

Recruitment period extended; Timeline and Methods updated.

Reviewer #2 Comment 5:

How long since the sexual assault happened? Because chronic symptoms are more difficult to respond. Have you thought about that?

Response:

We appreciate this insight. We have added a criterion that participants must have experienced the sexual assault at least 6 months prior to enrollment, to ensure stability of chronic PTSD symptoms and to avoid acute-phase interventions. This has been clarified under the “Eligibility Criteria” section.

Action Taken:

New eligibility criterion added: trauma must have occurred ≥6 months prior.

Reviewer #2 Comment 6:

For the sample size many studies show attrition rates of almost 30%, but you decided to calculate with 20%, maybe the sample size will lead to no significant result.

Response:

Thank you for highlighting this. While we initially used a 20% attrition estimate based on prior SE pilot data, we now note this limitation in the “Sample Size” section and acknowledge that higher attrition may affect power. We have also added a contingency plan in the Discussion to consider re-recruitment or per-protocol analysis if attrition exceeds 30%.

Action Taken:

Sample Size and Discussion sections updated with attrition caveats.

Reviewer #2 Comment 7:

The TAU that you mention as a comparison for the 10 SE sessions does not seem adequate, because is it an only 90 minutes session? and that is all the treatment they will receive for PTSD? The amount of time spent with the patients can be a serious confounding factor.

Response:

Thank you. We agree that this discrepancy may introduce bias. While the TAU reflects the real-world availability of services in our setting, we have clarified that it will consist of five group counseling sessions (90 minutes each), consistent with the average number of sessions reported in Indonesian practice and literature for survivors of sexual assault (ranging from 4 to 7 sessions across modalities such as Reality Counseling, EMDR, crisis counseling, and ACT).

It is important to emphasize that the purpose of this trial is to test the effectiveness of a culturally adapted SE intervention, rather than to determine which treatment is superior. Accordingly, TAU serves as a pragmatic benchmark condition that mirrors standard care in Indonesia, rather than a matched-dose comparator. While the unequal number of sessions may introduce a confounding factor, this is explicitly acknowledged as a limitation, and total professional contact time will be logged and controlled for in secondary analyses, with sensitivity models conducted to test robustness of the findings.

Action Taken:

Clarification added in Methods: TAU now described as five group counseling sessions (90 minutes each) based on evidence from Indonesian practice and literature.

Clarified in the Discussion under Limitations that TAU is intended as a real-world comparator, not a dose-matched control.

Explicit commitment to control for total intervention time in secondary and sensitivity analyses.

Reviewer #2 Comment 8:

I would suggest that you include some instrument to evaluate dissociation that I think is a very frequent symptom for sexual assault victims.

Response:

Thank you very much for this important suggestion. We fully agree that dissociation is a relevant symptom among sexual assault survivors. However, given the scope and focus of our current study, we decided to prioritize instruments that directly address PTSD symptoms, resilience, and quality of life, which are the core outcomes of our intervention. Including an additional measure of dissociation at this stage would risk overburdening participants with too many questionnaires and could potentially affect response quality. We therefore respectfully decided not to include an additional dissociation measure, but we acknowledge its importance and will consider it for future studies focusing more specifically on dissociation.

Action Taken:

No changes made. Justification provided above.

Yours sincerely, also on behalf of the co-authors,

Ligina Ayudia (corresponding author)

Faculty of Psychology

Universitas Padjadjaran, Indonesia

Jl. Raya Bandung Sumedang KM.21, Jatinangor, Jawa Barat, Indonesia

Email: ligina18001@mail.unpad.ac.id

---

## [Decision Letter · Decision Letter 1]

3 Nov 2025

Study Protocol for a Randomized Controlled Trial of Group-Adapted Somatic Experiencing®  Intervention for Indonesian Women Survivors of Sexual Assault with PTSD Symptom

PONE-D-25-39124R1

Dear Dr. Ligina Ayudia,

We’re pleased to inform you that your manuscript has been judged scientifically suitable for publication and will be formally accepted for publication once it meets all outstanding technical requirements.

Kind regards,

Fadwa Alhalaiqa

Academic Editor

PLOS ONE

Additional Editor Comments (optional):

Reviewers' comments:

Reviewer's Responses to Questions

**Comments to the Author**

1. Does the manuscript provide a valid rationale for the proposed study, with clearly identified and justified research questions?

Reviewer #1: Yes

2. Is the protocol technically sound and planned in a manner that will lead to a meaningful outcome and allow testing the stated hypotheses?

Reviewer #1: Yes

3. Is the methodology feasible and described in sufficient detail to allow the work to be replicable?

Reviewer #1: Yes

4. Have the authors described where all data underlying the findings will be made available when the study is complete?

Reviewer #1: Yes

5. Is the manuscript presented in an intelligible fashion and written in standard English?

Reviewer #1: Yes

You may also provide optional suggestions and comments to authors that they might find helpful in planning their study.

Reviewer #1: All concerns have been addressed.

XXXXXXXXXXXXXXXXXXXXXXXXXXXXXXXXXXXXXXXXXXXXXXXXXXXXXXXXXXXXXXXXXXXXXXXXXXXXXXXXXXXXXXXXXXXXXXXXXXXXXXXXXXXXXXXXXXXXXXXXXXXXXXXXx

**Do you want your identity to be public for this peer review?** For information about this choice, including consent withdrawal, please see our Privacy Policy

Reviewer #1: No

---

## [Editor Report · Acceptance letter]

PONE-D-25-39124R1

PLOS ONE

Dear Dr. Ayudia,

I'm pleased to inform you that your manuscript has been deemed suitable for publication in PLOS ONE. Congratulations! Your manuscript is now being handed over to our production team.

Kind regards,

on behalf of

Pro Fadwa Alhalaiqa

Academic Editor

PLOS ONE